# Intradermal Delivery of Naked mRNA Vaccines via Iontophoresis

**DOI:** 10.3390/pharmaceutics15122678

**Published:** 2023-11-26

**Authors:** Mahadi Hasan, Anowara Khatun, Kentaro Kogure

**Affiliations:** 1Department of Animal Disease Model, Research Center for Experimental Modeling Human Disease, Kanazawa University, Kanazawa 920-8640, Japan; mahadihasan@kiea.m.kanazawa-u.ac.jp (M.H.); anowarakhatun@kiea.m.kanazawa-u.ac.jp (A.K.); 2Graduate School of Biomedical Sciences, Tokushima University, Tokushima 770-8505, Japan

**Keywords:** iontophoresis, intradermal delivery, mRNA vaccine, gap junction, endocytosis

## Abstract

Messenger RNA (mRNA) vaccines against infectious diseases and for anticancer immunotherapy have garnered considerable attention. Currently, mRNA vaccines encapsulated in lipid nanoparticles are administrated via intramuscular injection using a needle. However, such administration is associated with pain, needle phobia, and lack of patient compliance. Furthermore, side effects such as fever and anaphylaxis associated with the lipid nanoparticle components are also serious problems. Therefore, noninvasive, painless administration of mRNA vaccines that do not contain other problematic components is highly desirable. Antigen-presenting cells reside in the epidermis and dermis, making the skin an attractive vaccination site. Iontophoresis (ItP) uses weak electric current applied to the skin surface and offers a noninvasive permeation technology that enables intradermal delivery of hydrophilic and ionic substances. ItP-mediated intradermal delivery of biological macromolecules has also been studied. Herein, we review the literature on the use of ItP technology for intradermal delivery of naked mRNA vaccines which is expected to overcome the challenges associated with mRNA vaccination. In addition to the physical mechanism, we discuss novel biological mechanisms of iontophoresis, particularly ItP-mediated opening of the skin barriers and the intracellular uptake pathway, and how the combined mechanisms can allow for effective intradermal delivery of mRNA vaccines.

## 1. Introduction

Since the COVID-19 pandemic, remarkable progress has been made in developing messenger RNA (mRNA) vaccines. The United States Food and Drug Administration (FDA) approved two mRNA vaccines (mRNA-1273 and BNT 162b2) that have shown great potential to combat COVID-19 [1,2]. After the success in controlling COVID-19, the development of mRNA-based personalized cancer vaccines started to attract considerable attention [3,4]. More than twenty mRNA-based anticancer immunotherapies are currently being investigated in clinical trials [5]. mRNAs are macromolecules that contain information to synthesize proteins. In the cell, ribosomes read mRNA macromolecules and translate their sequences into proteins in the cytoplasm [6]. The principle of the mRNA vaccination platform is to deliver a synthetic mRNA macromolecule encoding the desired protein or polypeptide antigen to induce humoral or cell-mediated immune responses that will be effective against a pathogen or malignant cells. mRNA vaccines offer several advantages over conventional vaccines (e.g., peptide and virus-vector-based vaccines) and DNA-based vaccines [7,8,9,10]. For example, mRNA produces antigens following one-step translation in cells, so that the rate and expression ability of an mRNA vaccine’s antigen is higher than that of a DNA vaccine [7]. Additionally, multiple antigens can be encoded by a single mRNA sequence, which maximizes the immune response against malignant cells or resilient pathogens [11]. In contrast to DNA vaccines, the active site of the mRNA vaccines is the cell cytoplasm; as such, there is no risk for integration into the host genome [12]. Moreover, the manufacturing process for mRNA vaccines is rapid and inexpensive, and the vaccines can be produced in a cell-free manner [13]. On the other hand, mRNA vaccines also exhibit some limitations, such as high molecular weights, low stability, and being negatively charged [7]. mRNA macromolecules are also generally unable to cross the anionic plasma membrane of cells [14,15]. Moreover, mRNAs are degraded by the ribonucleases in living systems [16]. Recently, several modification technologies have been introduced for designing highly efficient mRNAs [17,18]. For example, the addition of a poly (A) tail has been shown to increase stability and translational efficiency; 5′-capping was shown to increases stability and protein synthesis; incorporation of a modified nucleoside was also shown to increases stability; and addition of an untranslated region (UTR) modulated the length, structure, and translation efficiency and prolonged the half-life of mRNA [19,20,21,22,23]. These methodological innovations markedly improve the stability of mRNA and support clinical applications.

In addition to mRNA design, consideration of the route of administration and effective delivery into the cytoplasm are also essential factors to achieve a desired therapeutic effect. However, efficient entry of mRNA molecules in the cells involves further challenges. mRNA molecules must overcome tissue, extracellular and intracellular barriers before they arrive at the targeted site. Following this journey, mRNAs are easily cleared by the immune system, degraded by nuclease in the extracellular environments, face repulsion from the plasma membrane, are trapped by endosomes, and can be degraded via intracellular immunity [24]. Therefore, the therapeutic amount of mRNA molecules in the cells is significantly reduced. To overcome this hurdle, mRNA vaccines need an effective delivery method. As the membrane permeation of mRNA is extremely low, a variety of delivery carriers containing lipids, lipid-like materials, polymers, and protein derivatives have been developed to deliver mRNA [25,26,27,28]. Lipid-based nanoparticles typically use cationic lipids or ionizing lipids as their main component [29]. Regardless of pH, cationic lipids (e.g., DOTMA and DOTAP) retain their cationic nature as they contain alkylated quaternary ammonium groups. Conversely, ionizing lipids (e.g., ATX-100 and LP-01) become protonated in free amines at low pH and exhibit a positive charge. Lipid-like materials (e.g., N1,N3,N5-tris(2-aminoethyl)benzene-1,3,5-tricarboxamide (TT) derivatives) contain more hydrophobic side chains compared to neutral lipids [28,29,30]. These carriers encapsulate mRNA and protect from enzymatic degradation, facilitating cellular uptake and endosomal escape. Besides lipid or lipid-like nanocarriers, polymeric materials (e.g., PEI and PLGA) and cell-penetrating peptides (e.g., TAT and RALA) have been investigated for mRNA delivery [27,31]. However, considering the issue of biocompatibility of the polymeric materials and the mechanism of cell-penetrating peptides, they still need to be clinically advanced. To date, lipid-based nanoparticles have been widely investigated and used in clinical applications among these nanocarriers. However, ionizing cationic lipids are the main component of lipid-based nanoparticles, and have been shown to cause acute side effects such as pain, fever, swelling, and an immune response that can lead to anaphylactic-like shock [32,33,34]. Therefore, development of mRNA vaccines without the need for a problematic carrier or other components is highly desirable. Recently, intranodal and intradermal administration of naked mRNA has been investigated [35,36]. Sebastian et al. reported the potent immune response upon delivery of a single epitope via the intranodal injection of naked mRNA using murine models [35]. Furthermore, Sonia et al. investigated the intradermal delivery of naked mRNA encoding a fluorescent protein into excised pig skin [36]. They found that the intradermal delivery of naked mRNA resulted in protein expression. Although these findings did not discuss some points (e.g., clinical application, doses of mRNA, and expression amount), they indicate proof of concept and feasibility of the application of naked mRNA. Among these routes of administration, intradermal administration is preferable over intranodal administration, as the latter is difficult to perform, while the dermis is rich in antigen-presenting cells (APCs) [37,38]. However, intradermal administration has typically been carried out using invasive technique, such as injection with a needle, which causes pain, swelling, and risk of infection. Moreover, a trained person is required to perform the injection [39,40,41,42]. To overcome these limitations, various physical methods such as iontophoresis (ItP), sonophoresis, electroporation, microinjection, and pyro jet injector have been investigated to enable noninvasive or minimal invasive intradermal delivery of macromolecules [43]. Among these physical methods, we focus on ItP technology for noninvasive intradermal administration of naked mRNA vaccines.

ItP is a noninvasive intradermal permeation technology that uses weak electric current (WEC) applied to the surface of the skin [44,45,46,47]. The electricity used in ItP is ≤0.5 mA/cm^2^ [48]. The applied electric current can induce several events on the skin, such as electrostatic repulsion, electro-osmosis, cleavage of the intercellular junctions, and activation of the intracellular uptake pathway [49,50]. Cumulatively, these ItP-mediated mechanisms facilitate the intradermal permeation of macromolecules across the skin barriers and induce cytoplasmic delivery into skin cells. Herein, we highlight ItP technology over other delivery methods because of two specific advantages. First, ItP technology and the associated mechanisms can be beneficial for the effective delivery of naked mRNA vaccines utilizing these mechanisms. Second, ItP technology is inexpensive and does not require complicated devices or instrumental procedures compared to other methods. Therefore, it is expected that ItP technology will reduce the burden of needle phobia and improve patient compliance. In this review, we discuss ItP-mediated intradermal delivery of several macromolecules, the mechanisms of ItP and advantages for intradermal naked mRNA delivery, highlight a recent example of ItP-mediated intradermal delivery of mRNA vaccines, and provide insights into outcomes and implications.

## 2. Intradermal Delivery of Various Macromolecules by ItP

ItP is generally thought to be suitable for transdermal delivery of small, hydrophilic, and charged molecules [51]. However, ItP has also been extensively studied over the past decade for intradermal delivery of naked and liposome-encapsulated macromolecules [49,50]. These studies are described below and summarized in Table 1. Kigasawa K. et al. first reported intradermal delivery of short-interfering RNA (siRNA) following ItP [52]. The authors observed a fluorescent signal up to the epidermal and dermal junction while ItP of fluorescently labeled siRNA was performed on the atopic dermatitis model rat skin. Moreover, the expression level of the interleukin-10 (IL-10) mRNA, a characteristic feature of atopic dermatitis [53], was significantly suppressed following the ItP of IL-10 siRNA. These results demonstrated effective delivery of siRNA into the skin cells via ItP.

Jose et al. investigated the intradermal delivery of liposomes containing signal transducer and activator of transcription 3 (STAT3) siRNA and curcumin for the treatment of melanoma [54]. Suppression of STAT3 expression reduced cancer progression, while curcumin also exhibited anticancer activity [55,56]. ItP of such liposomes on melanoma-bearing mouse skin suppressed cancer progression compared with curcumin or STAT3 liposomes alone. These results demonstrated that ItP-mediated intradermal co-delivery of siRNA and small molecular anticancer drugs can be a useful treatment against skin disease.

Aside from intradermal delivery, Hasan et al. evaluated the ItP of heat shock protein (HSP) 47 siRNA in a CCl_4_-induced fibrosis mouse liver, ItP of resistin siRNA in the liver of KK-A^y^ obesity model mice, and ItP of pancreatic and duodenal homeobox (Pdx)-1 siRNA on the pancreas of BALB/c mice. The authors found significant suppression of the corresponding mRNA expression in each of the models [44]. In addition to the siRNA, the ItP of peptides, enzymes, and antibodies have also been widely studied. Kigasawa et al. investigated the intradermal delivery of the antioxidative enzyme superoxide dismutase (SOD) for protection against UV-induced skin damage [57]. Skin is exposed to ultraviolet rays from solar radiation, which is associated with the generation of reactive oxygen species (ROS) that subsequently cause inflammation, the production of melanin, destruction of skin fiber, and alteration of DNA in the skin [58,59,60]. The authors formulated a cationic-liposome-encapsulating SOD for intradermal delivery of SOD to protect the UV-induced, above-skin damage. ItP of SOD liposomes on UVA-irradiated and 8-methoxypsoralen-treated rat skin significantly suppressed membrane damage-related markers. These results demonstrated that the ItP of SOD liposomes enabled intradermal delivery of the antioxidative enzyme SOD, preventing the generation of various types of oxidative products in the skin.

Kajimoto et al. investigated the intradermal delivery of insulin [61]. To increase the stability of the insulin, the authors first developed a cationic liposomal formulation to encapsulate insulin. Following ItP of insulin-encapsulated liposomes on rat skin, insulin was observed in the epidermis region. Further, ItP of the liposomes on the diabetic rats gradually reduced blood glucose levels, with the hypoglycemic effect maintained for up to 24 h. This study, thus, demonstrated the successful intradermal delivery of liposome encapsulated insulin by ItP.

Cyclosporin A is a cyclic peptide containing 11 amino acids [62]. Cyclosporin A is used as an immunosuppressant drug [62]. Intradermal delivery of cyclosporin A has excellent potential to treat various skin diseases [63,64]. Boinpally et al. investigated ItP-mediated delivery of lecithin liposomes encapsulating cyclosporin A across the epidermis of a human cadaver [65]. Although passive diffusion did not induce permeation of cyclosporin A, use of ItP-mediated epidermal delivery of cyclosporin A (230 μg), demonstrated the utility of combining ItP with cyclosporin A for topical immunosuppression.

Psoriasis is an immune-mediated chronic inflammatory disease that affects the skin [66]. Psoriatic skin is rich in immune cells such as T-cells, macrophages, neutrophils, NK cells, and dendritic cells [67,68,69]. These cells secrete tumor necrosis factor α (TNF-α), interleukin-6, and other pro-inflammatory cytokines that mediate the pathogenesis of psoriasis [67,68,69]. Fukuta et al. reported the ItP-mediated intradermal delivery of antibodies in mouse skin [70]. The authors found that the repetitive application of ItP of TNF-α drug etanercept (recombinant human TNF-α receptor: Fc fusion protein) on imiquimod (IMQ)-induced psoriasis skin significantly reduced the pathogenesis of psoriasis. Consequently, Lapteva et al. investigated the ItP-mediated skin delivery of cetuximab, a recombinant human/mouse chimeric monoclonal antibody against epidermal growth factor receptor (EGFR) [71]. EGFR is overexpressed in squamous cell carcinoma (SCC), and cetuximab is used to treat SCC [72]. The authors found that the ItP delivered a therapeutic concentration of cetuximab into the viable epidermis after 1 h, after 4 h for the upper dermis, and after 8 h for the lower dermis. These results suggest that ItP may be useful for efficient antibody delivery into the skin.

**Table 1 pharmaceutics-15-02678-t001:** Summary of ItP-mediated delivery of various macromolecules.

Author	Macromolecules	Study Model & and Dose of IP	Outcome
Kigasawa et al. [52]	IL-10 siRNA	Atopic dermatitis rat; 0.3 mA/cm^2^, for 1 h	ItP-induced intradermal delivery of siRNA and reduced IL-10 mRNA expression.
Hasan et al. [44]	HSP47 siRNA	CCl_4_-induced fibrosis mice; 0.34 mA/cm^2^, 30 min	ItP-induced hepatic delivery of siRNA and significantly suppressed HSP 47 mRNA expression as well as pathogenesis of fibrosis.
Hasan et al. [44]	Resistin siRNA	KKA^y^ obesity model mice; 0.34 mA/cm^2^, 30 min	ItP of siRNA significantly suppressed resistin expression and hepatic lipid accumulation.
Hasan et al. [44]	Pdx-1 siRNA	BALB/c mice, 0.34 mA/cm^2^, 30 min	Significantly suppressed Pdx-1 expression in pancreas.
Jose et al. [54]	Liposome-encapsulated STAT3 siRNA and curcumin	Murine model of melanoma; 0.47 mA/cm^2^, 2 h	ItP-mediated co-delivery showed potent tumor suppression relative to single applications.
Kigasawa et al. [57]	Liposomal superoxide dismutase	UV-irradiated Rats; 0.45 mA/cm^2^, 1 h	ItP-mediated delivery of SOD reduced UV-induced skin damage.
Kajimoto et al. [61]	Liposome-encapsulated insulin	Diabetic rats 0.45 mA/cm^2^, 1 h	Stably maintained the blood glucose level up to 24 h.
Boinpally et al. [65]	Liposomes-encapsulate cyclosporin A	Human cadaver epidermis (in vitro); 0.57 mA/cm^2^, 5 h	ItP-mediated intradermal delivery of cyclosporin A will be useful for local immunosuppression.
Fukuta et al. [70]	TNF-α drugetanercept	IMQ-induced psoriasis rat; 0.34 mA/cm^2^, 1 h	ItP significantly reduced the epidermal hyperplasia and pathogenesis of psoriasis.
Lapteva et al. [71]	Cetuximab	Porcine skin; 0.5 mA/cm^2^, 2, 4, 8 h	ItP-employed intradermal delivery of cetuximab.

From this section, we observed that ItP technology enabled the intradermal delivery of various macromolecules. Most of these applications showed effectiveness against skin-associated diseases, such as ItP of IL-10 for atopic dermatitis, STAT3 siRNA for skin cancer, and TNF-α drug etanercept for psoriasis. However, the application of ItP for intradermal delivery of macromolecules remains at the pre-clinical stage and has not yet been clinically used. In addition to macromolecules, ItP of small molecule drugs is investigated for the treatment of a wide range of skin conditions. This manuscript does not review this context as we focus on ItP-mediated intradermal delivery of macromolecules and mRNA vaccination. For the application of ItP in dermato-cosmetic and aesthetic sciences, interested readers can follow the review with reference [73].

## 3. Mechanisms of ItP and Its Advantages for Intradermal Delivery of Naked mRNA Vaccines

ItP can facilitate intradermal delivery of small-molecule drugs and large biological macromolecules. However, it may not be clear just how these molecules are able to permeate across the skin barrier based on general electrochemical principles alone. In fact, the WEC applied during ItP also has an extensive effect on the skin barrier and skin residing cells. Therefore, in addition to the electrochemical mechanism, ItP is also associated with biological mechanisms that mediate the intradermal delivery of substances. The mechanisms associated with ItP are reviewed below and summarized in Figure 1.

### 3.1. Mechanism of ItP from an Electrochemical Perspective

ItP is performed by applying a WEC on the skin’s surface using two electrodes (the cathode and anode), an electric controller, a power source, and a drug reservoir [74]. During ItP, the electricity flows from the electrode to the skin. This flow of electricity is associated with two electrochemical phenomena, namely electrostatic repulsion and electro-osmosis [49,50,75]. Electrostatic repulsion occurs when two similarly charged particles come into close proximity and repel each other [48]. During ItP, under the influence of a positive or negative electrode, positively charged or negatively charged molecules, respectively, migrate across the skin barrier [48]. Electrostatic repulsion is, therefore, the direct effect of the application of an electric current to charged molecules. Small charged molecules can penetrate through the intercellular space between skin cells via electrostatic repulsion. Conversely, electro-osmosis is defined as the convective solvent flow under the influence of electricity [76]. The electromigration of charged entities induces electro-osmosis during the flow of current. The outermost layer of the skin, the stratum corneum (SC), is recognized as a physical barrier to skin permeation [77,78]. However, SC does not exhibit a regular structure, and the electric resistance of the SC is not consistent throughout the SC [79]. It contains several skin appendages (e.g., hair follicles, sweat glands, and sebaceous glands) [80]. These appendages extend from the skin surface to the dermis or lower dermis layers [81]. During ItP, electro-osmosis preferentially occurs following the trans-appendageal pathway. Consequently, electro-osmosis favors an anodal ItP of positively charged molecules and passive diffusion of neutral molecules [49,50].

### 3.2. Biological Mechanisms of ItP-Mediated Intradermal and Cytoplasmic Drug Delivery

The skin exhibits a distinct barrier function that restricts the entry of extraneous agents into the body. Skin barrier function is attributed to SC, which is composed of highly organized corneocytes surrounded by a lipid matrix containing ceramides, cholesterol, and fatty acids [82]. Underneath the SC is the viable epidermis, followed by the dermis layer. Topically applied drugs or macromolecules have three possible routes to overcome SC [83,84]. For example, the intracellular route through the cells, the intercellular or paracellular route between the cells, and the follicular or trans-appendageal route following hair follicles. Small lipophilic molecules can penetrate SC following the intracellular route. On the other hand, hydrophilic polar molecules are favorable for paracellular or trans-appendageal routes. However, penetration following these routes typically depends on the properties of drug molecules, the vehicle used in the formulation, and the types of drug delivery systems. Various intercellular junctions (e.g., gap junctions, tight junctions, and adherent junctions) are present between the cells of the skin [85,86]. These junctions maintain the integrity of the skin and the paracellular transport pathway to control the entry of small water-soluble molecules and pathogens into the skin [85,86]. Thus, these junctions act as a significant barrier to intradermal permeation. The structural proteins of the intercellular junctions are reported to be sensitive to chemical and physical stimuli [87]. Moreover, the structural protein of the gap junction connexin 43 in cardiac muscle cells was found to be responsive to an electric stimulus [88]. Based on this, Hama et al. evaluated the expression of connexin 43 following ItP with cationic liposomes [89]. The authors found that the application of ItP significantly reduced the levels of connexin 43, suggesting that ItP treatment cleaves the intercellular gap junctions of skin cells. The authors also observed that ItP treatment with liposomes increased the intracellular influx of Ca^2+^ and active protein kinase C (PKC). The increased cytoplasmic Ca^2+^ levels depolymerized the filamentous actin, which altered the properties of tight junctions. The activated PKC, on the other hand, increased phosphorylation of connexin 43, which could be the reason for the reduction in connexin 43 levels as well as the cleavage of the gap junctions. Consequently, Khatun et al. evaluated the effect of ItP on the gap junction proteins in B16F1 tumors using immunofluorescence analysis [90]. The authors found that ItP treatment significantly reduced the amount of connexin 43 by approximately 57% relative to control levels while it increased the amount of phosphorylated connexin 43 by approximately 25%. On the other hand, the amount of phosphorylated PKC was increased by approximately 28% relative to control levels in ItP-treated tumors. Taken together, these results suggest that the application of ItP leads to the activation of signaling molecule PKC, followed by the phosphorylation of connexin 43. The connexin 43 phosphorylation promotes the dissociation of gap junctions. Therefore, in addition to the appendageal pathway, ItP-mediated dissociation of the intercellular junctions also opens the paracellular transport pathway across the skin barriers.

RNA interference (RNAi) is a conserved biological process that takes places in the cytoplasm [91]. RNAi suppresses sequence-specific mRNA expression induced by siRNA. ItP of naked IL-10 siRNA or liposome-encapsulated STAT3 siRNA showed a significant RNAi effect on rat skin of an atopic dermatitis model or the skin of melanoma-bearing mice, respectively [52,54]. Taken together, these results demonstrated that siRNA was delivered into the cytoplasm of the corresponding cells after ItP-mediated permeation of the skin barriers.

Hasan et al. investigated the ItP-mediated cytoplasmic delivery and found that applying WEC with siRNA to culture cells showed significant RNAi effect, similar to in vivo ItP [92]. To visualize this cytoplasmic delivery, the authors applied WEC to the culture cells in the presence of in-stem molecular beacon (ISMB). ISMB is an oligonucleotide probe that fluoresces upon binding with the complementary mRNA in the cytoplasm. At 1 h after WEC of ISMB against luciferase, a potent fluorescence signal was observed in luciferase-expressing cells, while no fluorescence signal was observed in the absence of the WEC, nor after WEC with ISMB against GFP in luciferase-expressing cells. These results confirm that the WEC of ItP can induce cytoplasmic delivery of extraneous substances [92]. Various endocytosis inhibitors, such as low-temperature exposure, macropinocytosis inhibitor amiloride, and caveola-mediated endocytosis inhibitor filipin, were also found to significantly reduce WEC-mediated cytoplasmic delivery of fluorescently labeled siRNA. In parallel, immediately after WEC, the trypan blue exclusion test confirmed that the applied electric field did not induce electroporation or membrane damage to cells. Taken together, these results suggest that WEC-mediated cytoplasmic delivery occurs via endocytosis. Endocytosis was also visualized using confocal laser scanning microscopy (CLSM). After WEC of FITC-labeled DNA fragments (MW: 225,000), the endosomes were stained with LysoTracker Red. Green DNA was found to be co-localized with red endosomes in the cells via CLSM, indicating that after WEC the DNA remained in the endosomes [93]. For effective cytoplasmic delivery, the endosomal escape is essential, with escape efficiency typically dependent on the size of internalized molecules. Consequently, it was observed that the hydrophilic macromolecule FITC-Dextran 10,000 was widely distributed in the cytoplasm after 24 h of WEC, while FITC-Dextran 70,000 remained in the endosomes. These results demonstrated that the properties of endosomes are leaky after treatment with WEC, allowing for macromolecules exhibiting a molecular weight < 70,000 to escape [85].

To better understand the signaling pathway of WEC-based cellular uptake, Hasan et al. performed isobaric tags for relative and absolute quantification (iTRAQ) proteomic analysis and found that treatment of cells with WEC activates numerous downstream signaling molecules [94]. Several bioinformatics tools and Western blot analysis showed that PKCγ and HSP90α are the most interactive molecules in the WEC-based cellular uptake process. The combined activation of PKCγ and HSP90α following WEC subsequently activates Rho GTPase to induce actin cytoskeleton remodeling, leading to cellular uptake [86]. To investigate the morphological properties of endocytosis induced by WEC, Torao et al. applied WEC to polyethylene glycol (PEG)ylated gold nanoparticles (100 nm, −50 mV) and observed the cells using transmission electron microscopy (TEM) [95]. TEM observation detected several gold nanoparticles in the endosomes. However, the endosomes exhibited elliptical morphology, which differs from the conventional endosome’s spherical shape [96]. The widths and depths of the endosomes were approximately 100–200 nm and 500 nm, respectively. The sizes of the endosomes were smaller than those of macropinosomes (>1 μm) [97]. Moreover, it was found that ceremide is localized in WEC-generated endosomes [95]. Ceremide can produce pore or channel structures in the membranes [98]. Therefore, localization of ceremide may be the reason for the leaky properties and the tubular morphology of the endosomes. Further studies are needed to clarify the details of WEC-mediated ceremide pore formation. Taken together, WEC of ItP induces unique endocytosis that employs cytoplasmic delivery of extraneous substances. WEC-generated endosomes are tubular and can leak macromolecules with molecular weights <70,000. The precise signaling pathway of WEC-mediated unique endocytosis remains unknown.

To this end, during the application of ItP, the WEC induced electro-osmosis, electrostatic repulsion, opening of intercellular junctions, and activation of intracellular uptake pathway. These ItP-induced combined effects facilitate paracellular and trans-appendageal routes enabling intradermal delivery and activating unique endocytosis for intracellular uptake by skin cells.

## 4. Advantages of ItP for Intradermal Vaccination with Naked mRNA

Noninvasive intradermal delivery of naked mRNA is desirable for effective vaccination and improved patient compliance. mRNA requires a rational delivery system to overcome skin and cellular barriers to reach the cytoplasm of the targeted cells. ItP is a transdermal permeation technology associated with different penetration mechanisms [49,50]. In the present study, appealing features of ItP were demonstrated that will be advantageous for noninvasive intradermal delivery of a naked mRNA vaccine. These features are described as follows: First, ItP facilitates the intradermal permeation of various hydrophilic macromolecules. Further, negatively charged siRNA showed functionality in the epidermis and dermis regions following ItP. Unlike the intracellular route, ItP utilizes the appendageal pathway and dissociates intercellular junctions to open the paracellular transport pathway. Electrostatic repulsion or electro-osmosis in such a pathway employs intradermal delivery of hydrophilic macromolecules with an average penetration depth of >100 μm. Therefore, following ItP, macromolecules, such as mRNA can reach to the APCs in the skin layers. Second, mRNA must cross the plasma membrane and escape from endosomal entrapment to enter the targeted cell’s cytoplasm to translate into the desired protein antigen. However, naked mRNA cannot overcome these barriers due to their large molecular weight and negative charges. Thus, additional carriers are required to enable their cytoplasmic delivery. Moreover, during cellular entry of mRNA molecules via endocytosis, they would be trapped in endosomes. Endocytosis pathways typically consist of early endosomes, recycling endosomes, multivesicular bodies, late endosomes, and lysosomes [99]. Following this pathway, mRNA molecules recycle to the cell surface or fuse with lysosomes, which contain degradation enzymes. Thus, endosomal escape is essential to insert mRNA molecules into the cytoplasm. Recently, it has been confirmed that mRNA remains stable under the applied electric field of ItP and even under the high voltage of electroporation [100]. Furthermore, WEC of ItP activates the intracellular signaling pathway and induces rapid and homogeneous delivery of extraneous macromolecules into the cell cytoplasm [92,93,94]. WEC-induced cytoplasmic delivery occurs via unique endocytosis that forms leaky endosomes via localization of ceremide that produces channels or pores in the endosomes. Since ribonucleases degrade naked mRNA in the extracellular environment and inside endosomes, a prolonged stay in such an environment reduces the mRNA vaccine activity. Therefore, ItP-mediated rapid cellular uptake and subsequent escape from leaky endosomes will facilitate effective cytoplasmic delivery of mRNA. Moreover, homogeneous delivery of mRNA may induce potent immune-cell activation. Taken together, application of ItP will enable effective cytoplasmic delivery of naked mRNA in APCs within the skin. ItP-based vaccination may ultimately increase patient compliance, as it is noninvasive and exhibits a simple application procedure that does not require complicated devices.

## 5. ItP-Based Approaches for Effective Intradermal Vaccination

ItP technology has been investigated for intradermal delivery of mRNA and other prophylactic vaccines. These approaches are reviewed below and summarized in Table 2.

### 5.1. ItP-Mediated Intradermal Delivery of Naked mRNA Vaccine Targeting Melanoma

As noted earlier, mRNA vaccines have gained considerable attention as a potential cancer immunotherapy [106]. Vaccination with mRNA results in expression of tumor-associated or tumor-specific antigens in APCs that induce humoral or cell-mediated immune responses [106]. Husseini et al. investigated ItP-mediated intradermal delivery of a naked, minimal mRNA-based vaccine encoding tumor-associated human gp100_25–33_ (KVPRNQDWL) as an immunotherapy for melanoma [101]. The authors introduced a short poly (A) tail into the mRNA sequence that protects the mRNA from de-capping and degradation. Before applying the vaccine, ItP-mediated intradermal delivery was analyzed using a FITC-labeled oligonucleotide exhibiting a high molecular weight (MW: 19,800). ItP application was found to deliver FITC-labeled oligonucleotide homogeneously into the skin layer. The penetration depth of the oligonucleotide was 100 μm following ItP application, indicating its accumulation within the epidermal layer, which ranges in thickness from 100 to 200 μm. As antigen-presenting epidermal Langerhans cells (LCs) reside in this layer, oligonucleotides were taken up by the LCs after ItP for subsequent immune response. Based on these results, the authors applied ItP to a non-formulated naked minimal mRNA vaccine encoding the tumor-associated antigen (TAA) human gp100_25–33_ on melanoma-bearing mice and evaluated the therapeutic outcomes. ItP application showed significant tumor volume regression compared to the non-vaccinated mice. Furthermore, the tumor-regression potency was higher with ItP application compared with subcutaneous injection of the vaccine. A possible reason for such potent tumor regression with ItP may be attributed to the homogeneous delivery and activation of the intracellular uptake of vaccine molecules by ItP. To confirm the immune responses, the authors found that there was an elevation in mRNA expression levels of various cytokines, such as IFN-γ, TNF-α, and IL-12b, in both the skin and tumor tissues in the ItP-vaccinated group compared to the non-vaccinated group. The elevated cytokine levels contribute to the suppression of tumor growth. Furthermore, the infiltration of cytotoxic CD8^+^ T cells was also found in the tumor tissue, which is responsible for tumor clearance. The tumor growth inhibition by ItP of mRNA might be due to not only antigen derived immune response but also an adjuvant effect. Therefore, the ItP treatment of mRNA encoding non-related peptide should be examined in the future. Taken together, this study is the first to demonstrate the application of ItP for intradermal delivery of a naked mRNA vaccine as a potential immunotherapy for melanoma.

### 5.2. Intradermal Delivery of Polyplex Vaccine by ItP

Besides mRNA vaccines, ItP technology has also been studied for intradermal delivery of polyplex vaccines. Kigasawa et al. studied the ItP technology for intradermal delivery of CpG oligodeoxyribonucleotides (ODN) to induce an immune response in B16F1 melanoma-bearing mice [102]. CPG-ODN is a synthetic DNA molecule. It contains an unmethylated CpG motif, which is found in bacterial DNA [107]. Utilizing a toll-like receptor, B-cells, dendritic cells, and monocytes take up CpG-ODN to induce a potent immune response [108]. The authors found that ItP application facilitated intradermal delivery of CpG-ODN and increased the production of pro-inflammatory cytokines as well as suppressed tumor growth.

Toyoda et al. studied the ItP-mediated intradermal delivery of a TAA gp100-loaded nanogel [103]. ItP application delivered the TAA gp100 into the epidermis and subsequently activated the LCs, resulting in the suppression of melanoma growth. Based on the results of these studies, Husseini et al. developed a polyplex vaccine containing the TAA peptide gp100_25–33_ (KVPRNQDWL-RRRR) and a negatively charged CpG-ODN adjuvant [104]. ItP of the vaccine on melanoma-bearing mice induced the homogeneous distribution of the vaccine into the skin. Further, a potent antitumor effect was observed, which was mediated by increased cytokine expression, especially interferon (IFN)-γ, as well as infiltration of cytotoxic CD8^+^ T cells. Finally, the dose-dependent ItP administration showed a significant reduction in tumor burden.

### 5.3. Improved Transdermal Delivery of Rabies Vaccines via ItP Coupled with Microneedles

ItP technology can be combined with other permeation methods, such as electroporation, ultrasound, and microneedle approaches [109,110,111]. The combined application can induce a cooperative effect that improves the delivery efficacy of biological macromolecules. Arshad et al. studied the transdermal delivery of rabies vaccines using ItP technology coupled with microneedle approaches [105]. This study demonstrated that the application of ItP in combination with microneedles provided a relatively potent immunogenic response over conventional intramuscular injection. Therefore, ItP technology, combined with other permeation methods, can also be useful for effective vaccination. ItP, in combination with other permeation methods, did not report any potential adverse effects. However, the changes in the skin structure, irritation, and erythema associated with the combined application of ItP with each method need to be investigated. Additionally, the combined application will be complicated and expensive relative to the single application of ItP.

## 6. Clinical Implication of ItP and Associated Trials

ItP has shown effectiveness in the intradermal delivery of a number of small molecule drugs against various skin disorders. Recently, ItP-mediated intradermal delivery of macromolecules and vaccines for the treatment of skin cancers has been investigated [49,50]. In addition to the dermal and intradermal application, ocular ItP and ItP on the internal organs have been explored [44,112]. Transpapillary, ItP-mediated delivery of chemotherapeutic drugs to specific cancer tissue to minimize systemic toxicity has been investigated [113]. Despite this wide range of applications, ItP is not clinically advanced, and most of its applications remain at the laboratory level. To date, FDA has approved ItP for dermal analgesia, management of post-operative pain, and migraines [45,114]. After surgery, patients suffer from post-operative pain. Conventionally, intravenously administrated, patient-controlled analgesia is used to manage post-operative pain. IONSYS^®^ (The Medicines Company, Parsippany, NJ, USA) was a needle-free fentanyl ItP patch that offers patient-controlled, on-demand fentanyl delivery to the post-operative patient and is used as an alternative to conventional pain management system [114]. Zecuity^®^ (Nupathe Inc., Conshohocken, PA, USA) was a sumatriptan ItP patch used for the management of migraines. Sumatriptan is a hydrophilic drug that acts as an anti-migraine agent [45]. Oral formulations of sumatriptan exhibits poor bioavailability. Moreover, a migraine attack is associated with nausea and vomiting, which makes oral administration unstable. Thus, ItP offers noninvasive transdermal delivery of sumatriptan that improves patient compliance. LidoSite^®^ (Vyteris Inc., Fair Lawn, NJ, USA) was an ItP patch that contained lidocaine and epinephrine [114]. After ItP-mediated combined delivery, epinephrine reduced the blood flow at the delivery site to increase the local concentration of lidocaine. Thus, the combined effect causes rapid dermal analgesia. Although FDA has approved these ItP patches, these products have been discontinued from the market because of some issues, such as the availability of the ItP device, safety, and lack of sales [45,114].

Recently, several clinical trials associated with ItP have been reported, for example, ItP of dexamethasone and gel lidocaine for the treatment of lateral epicondylitis [115], ItP of methotrexate against palmar psoriasis [116], ItP-mediated transdermal delivery of vitamin C for the treatment of photoaging skin [117], and ItP of neostigmine/glycopyrrolate to initiate bowel evacuation in patients with spinal cord injury [118]. These clinical trials showed promising results for treating corresponding diseases and ensuring the ItP approach is safe and effective. In recent years, the field of biological macromolecular drugs is expanding. Among them, mRNA vaccine development draws significant attention. Conventional hypodermic injection is used to administer the vaccines. We discussed the limitations of such administration earlier in this manuscript, which need to be improved. ItP of naked mRNA vaccines would be an alternative to conventional delivery methods. Furthermore, painless administration and easy application of ItP will enhance patient compliance.

## 7. ItP Instrument, Stability, and Storage of mRNA Vaccines

A primary ItP instrument typically contains a power source, an electric controller, two electrodes (e.g., anode and cathode), and a drug reservoir. Two electrodes are placed on the skin surface during ItP application to complete the circuit. The drug reservoir is embedded using the active electrode, while the return electrode contains counter ions. Earlier, the ItP device was large and required an external power source. However, due to the rapid development of the technology, the ItP device has been updated. Recently, self-powered, smart ItP devices have been introduced and are commercially available. Tula^®^ System (Tusker Medical Inc., Menlo Park, CA, USA), Hidrex PSP1000 (Hidrex USA, Austin, TX, USA), Idromed 5 PS (IontoCentre, Norwich, UK), and Fischer Galvanic MD-2 (RA Fischer Co., Moorpark, CA, USA) are some FDA-approved ItP devices available in the market [45,114].

mRNA vaccines showed remarkable success in combating the COVID-19 pandemic. Despite the effectiveness and safety of mRNA vaccines, their stability is not clearly understood. Several factors, such as the vehicle used in the formulation, pH, and temperature, can affect the stability of the mRNA vaccines [119]. Pfizer/BioNTech and Moderna used lipid nanoparticle-formulated mRNA vaccines. According to the European Medical Agency (EMA), both vaccines are stable in frozen conditions for up to 6 months at −25 °C, up to 30 days at refrigerator temperature (4 °C), and up to 6 h at room temperature [119]. Regarding the storage conditions of naked mRNA, no specific guideline is found. However, Jones et al. found that freeze-drying using trehalose permitted stable storage of functional RNA at 4 °C for up to 10 months [120].

## 8. Conclusions and Future Perspective

The development of mRNA vaccines against various infectious diseases and as anticancer immunotherapies has been widely studied in recent years. However, mRNA vaccines have typically relied on lipid nanoparticles as delivery vehicles, which are associated with various side effects. Furthermore, vaccine administration methods remain invasive and painful. In the present study, we reviewed the application of ItP technology for noninvasive intradermal delivery of naked mRNA vaccines. We discussed various ItP-associated mechanisms and their advantages for naked mRNA vaccination. Furthermore, we highlighted some ItP-mediated vaccination approaches that demonstrated effective immune activations. Taken together, ItP technology offers noninvasive, painless intradermal vaccination. Several challenges need to be overcome to extend the application of ItP technology for intradermal vaccination. The delivery efficacy of mRNA and associated immune responses need to improve. ItP patches equipped with mRNA vaccines and cost-effective smart ItP devices need to be developed. Due to the rapid development of microelectronics, cell-phone-used, remote-controllable dosage management ItP systems; self-powered ItP patches; and eco-friendly energy-used ItP patches have been reported [121,122,123]. These developments are expected to extend the applicability of ItP. Furthermore, for clinical translation of ItP technology, the use of human-relevant models such as human skin explants, tissue-engineered skin models, and human-based tissues grafted onto mice needs to increase [124,125,126]. More research and clinical trials will improve the application of ItP for vaccine delivery in the future.

## Figures and Tables

**Figure 1 pharmaceutics-15-02678-f001:**
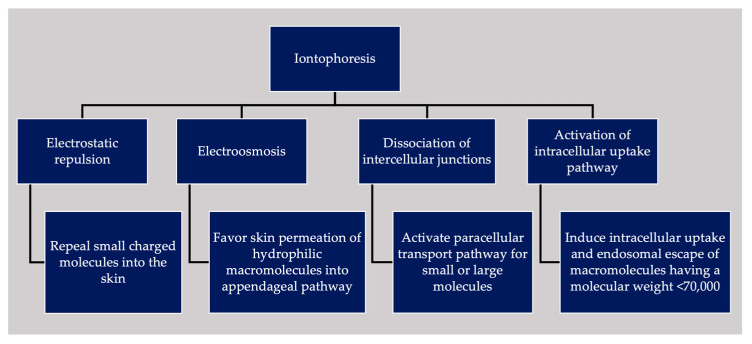
Summary of various ItP-induced permeation mechanisms and their outcomes.

**Table 2 pharmaceutics-15-02678-t002:** Brief summary of ItP-mediated intradermal vaccination approaches and their respective outcomes.

Author	Vaccine Components/Vaccine	Study Model and Dose of IP	Outcome
Husseini et al. [101]	Naked mRNA encoding tumor-associated antigen gp100_25–33_	Melanoma-bearing mice; 0.17 mA/0.5 cm^2^ for 1 h.	ItP caused significant regression in the tumor volume via cytokine production and activation of cytotoxic CD8^+^ T cells.
Kigasawa et al. [102]	Naked CpG-ODN	Melanoma-bearing mice; 0.3 mA/cm^2^, for 1 h	ItP induced pro-inflammatory cytokine production and suppressed tumor growth.
Toyoda et al. [103]	Tumor-associated antigen gp100 loaded nano gel	Melanoma-bearing mice; 0.4 mA/cm^2^, 1 h	ItP application activated immune responses and suppressed tumor growth.
Husseini et al. [104]	Polyplex containing gp100_25–33_ and CpG-ODN	Melanoma-bearing mice; 0.34 mA/cm^2^ for 1 h	ItP employs a potent antitumor effect via cytokine production and activation of cytotoxic CD8^+^ T cells.
Arshad et al. [105]	Rabies vaccine	Used beagle dogs; 0.5 mA/cm^2^ for 10 min couple with microneedle	ItP activated potent immunogenic response relative to the conventional intramuscular injection.

## Data Availability

Not applicable.

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
