# Peer review of "Intradermal Delivery of Naked mRNA Vaccines via Iontophoresis"

_pharmaceutics, 2023, doi:10.3390/pharmaceutics15122678_

Round 1

Reviewer 1 Report

Comments and Suggestions for Authors

Attached

Comments on the Quality of English Language

Moderate editing of English language required

Author Response

To Reviewer#1

The aim of this review is to explore the potential of iontophoresis (ItP) as a non-invasive, painless method for intradermal delivery of naked mRNA vaccines. By addressing the challenges associated with conventional mRNA vaccination, such as pain, needle phobia, and side effects, the review aims to highlight the advantages and mechanisms of ItP technology in facilitating effective mRNA vaccine delivery. Although the review presents a certain scientific interest, there are some concerns, and here are some important comments that will help readers understand the practical significance of these studies:

<Response to Reviewer#1’s comment>

Thank you for reviewing our manuscript. We wish to express our appreciation to Reviewer#1 for valuable comments.

<Reviewer#1’s specific comment#1>

Elaborate further on the limitations of mRNA vaccines, explaining why they are significant and how they relate to the need for effective delivery methods.

<Response to Reviewer#1’s specific comment#1>

Based on the reviewer's comment, we further elaborated on the limitations of mRNA vaccines and added additional texts in the revised manuscript as follows:

However, efficient entry of mRNA molecules in the cells involves further challenges. mRNA molecules must overcome tissue and extracellular and intracellular barriers before they arrive at the targeted site. Following this journey, mRNAs are easily cleared by the immune system, degraded by nuclease in the extracellular environments, face repulsion from the plasma membrane, are trapped by endosomes, and can be degraded by intracellular immunity [24]. Therefore, the therapeutic amount of mRNA molecules in the cells is significantly reduced. To overcome this hurdle, mRNA vaccines need an effective delivery method. (Page ##2, Lines ##64-71 in the revised manuscript)

Reference:

[24] Pilkington, E.H.; Suys, E.J.A.; Trevaskis, N.L.; Wheatley, A.K.; Zukancic, D.; Algarni, A.; Al-Wassiti, H.; Davis, T.P.; Pouton, C.W.; Kent, S.J.; et al. From Influenza to COVID-19: lipid nanoparticle mRNA vaccines at the frontiers of infectious diseases. Acta Biomaterialia 2021, 131, 16–40. doi: https://doi.org/10.1016/j.actbio.2021.06.023.

<Reviewer#1’s specific comment#2>

Emphasize the clinical relevance of ItP as a non-invasive method for intradermal mRNA vaccine delivery, highlighting specific advantages over other delivery methods.

<Response to Reviewer#1’s specific comment#2>

Based on the review’s comment, we added some additional explanations in the revised manuscript as follows:

To overcome these limitations, various physical methods such as iontophoresis (ItP), sonophoresis, electroporation, microinjection, and pyro jet injector have been investigated to enable noninvasive or minimal invasive intradermal delivery of macromolecules [43]. Among these physical methods, we focus on ItP technology for noninvasive intradermal administration of naked mRNA vaccines. (Page ##3, Lines ##103-108 in the revised manuscript)   

Herein, we highlight ItP technology over other delivery methods because of two specific advantages. First, ItP technology and the associated mechanisms can be beneficial for the effective delivery of naked mRNA vaccines utilizing these mechanisms. Second, ItP technology is inexpensive and does not require complicated devices or instrumental procedures compared to other methods. Therefore, it is expected that ItP technology will reduce the burden of needle phobia and improve patient compliance. (Page ##3, Lines ##115-121 in the revised manuscript)

Reference:

[43] Zhang, H.; Pan, Y.; Hou, Y.; Li, M.; Deng, J.; Wang, B.; Hao, S. Smart physical based transdermal drug delivery system: towards intelligence and controlled release. Small 2023, doi: https://doi.org/10.1002/smll.202306944.

<Reviewer#1’s specific comment#3>

Please, ensure that the promised discussion of recent examples of ItP-mediated intradermal delivery of mRNA vaccines is included in later sections of the paper, providing insights into outcomes and implications.

<Response to Reviewer#1’s specific comment#3>

Although ItP technology is investigated for the intradermal delivery of various macromolecules, to date Husseini et al.  (Biol. Pharm. Bull. 2023, 46, 301–308) is the only study to report the ItP-mediated intradermal delivery of mRNA vaccine. We discussed this study, containing its outcome and implication in section 5.1. and it is summarized in Table 4. Based on the reviewer's comment, we added the following text at the end of the introduction section to ensure it.

highlight a recent example of ItP-mediated intradermal delivery of mRNA vaccines, providing insights into outcomes and implications. (Page ##3, Lines ##123-124 in the revised manuscript)

<Reviewer#1’s specific comment#4>

Please, provide a brief overview of the types of delivery carriers mentioned in the introduction and their pros and cons, leading into the discussion of ItP as an alternative.

<Response to Reviewer#1’s specific comment#4>

According to the reviewer's comment, we added a brief overview of the types of delivery carriers in the revised manuscript as follows.

Lipid-based nanoparticles typically use cationic lipids or ionizing lipids as their main component [29]. Regardless of pH, cationic lipids (e.g., DOTMA,  DOTAP) retain their cationic nature as they contain alkylated quaternary ammonium groups. Conversely, ionizing lipids (e.g., ATX-100, LP-01) become protonated in free amines at low pH and exhibit a positive charge. Lipid-like materials (e.g., N1,N3,N5-tris(2-aminoethyl)benzene-1,3,5-tricarboxamide (TT) derivatives) contain more hydrophobic side chains compared to neutral lipids [28-30]. These carriers encapsulate mRNA and protect from enzymatic degradation, facilitating cellular uptake and endosomal escape. Besides lipid or lipid-like nanocarriers, polymeric materials (e.g., PEI, PLGA) and cell-penetrating peptides (e.g., TAT, RALA) have been investigated for mRNA delivery [27,31]. However, considering the issue of biocompatibility of the polymeric materials and the mechanism of cell-penetrating peptides, they still need to be clinically advanced. To date, lipid-based nanoparticles have been widely investigated and used in clinical applications among these nanocarriers. (Page ##2, Lines ##73-86 in the revised manuscript)

Reference:

[29] Zhang, W.; Jiang, Y.; He, Y.; Boucetta, H.; Wu, J.; Chen, Z.; He, W. Lipid carriers for mRNA delivery. Acta Pharm. Sin. B 2022, 13, 4105-4126, doi: https://doi.org/10.1016/j.apsb.2022.11.026.

[28] Kowalski, P.S.; Rudra, A.; Miao, L.; Anderson, D.G. Delivering the messenger: advances in technologies for therapeutic mRNA Delivery. Mol. Ther. 2019, 27, 710–728, doi: https://doi.org/10.1016/j.ymthe.2019.02.012.

[30] Zhang, X.; Zhao, W.; Nguyen, G.N.; Zhang, C.; Zeng, C.; Yan, J.; Du, S.; Hou, X.; Li, W.; Jiang, J.; et al. Functionalized lipid-like nanoparticles for in vivo mRNA delivery and base editing. Sci. Adv. 2020, 6, eabc2315, doi: https://doi.org/10.1126/sciadv.abc2315.

[27] Yang, W.; Mixich, L.; Boonstra, E.; Cabral, H. Polymer‐Based MRNA Delivery Strategies for Advanced Therapies. Adv. Healthc. Mater. 2023, 12, e2202688, doi:https://doi.org/10.1002/adhm.202202688.

[31] Kim, Y.; Kim, H.; Kim, E.H.; Jang, H.; Jang, Y.; Chi, S.-G.; Yang, Y.; Kim, S.H. The potential of cell-penetrating peptides for mRNA delivery to cancer cells. Pharmaceutics 202214, 1271, doi: https://doi.org/10.3390/pharmaceutics14061271.

<Reviewer#1’s specific comment#5>

Please, explain in detail how the use of ItP enhanced macromolecule delivery and the observed effects on target tissues or diseases.

<Response to Reviewer#1’s specific comment#5>

ItP is a noninvasive transdermal permeation technology that uses weak electric current (WEC) on the skin surface. The use of ItP on the skin surface has two types of mechanisms or effects that enhance macromolecules delivery into the skin. First, the applied WEC of ItP provides a driving force that induces electroosmosis and electrostatic repulsion of macromolecules towards target skin or tissue in normal or disease conditions. Second, the WEC has some biological effects on the applied skin or tissue surface, such as disrupting the intercellular junctions and activating intracellular signaling pathways. During the use of ItP, these above mechanisms or effects worked together to enhance macromolecule delivery in normal or disease conditions. We already discuss the detailed mechanism of ItP and how it facilitates macromolecule delivery in section 3. Furthermore, we added a summary at the end of section 3 in the revised manuscript as follows:

To this end, during the application of ItP, the WEC induced electroosmosis, electrostatic repulsion, opening of intercellular junctions, and activation of intracellular uptake pathway. These ItP-induced combined effects facilitate paracellular and trans-appendageal routes enabling intradermal delivery and activating unique endocytosis for intracellular uptake by skin cells. (Page ##9, Lines ##325-329 in the revised manuscript)

<Reviewer#1’s specific comment#6>

Please, highlight the clinical relevance of ItP for intradermal macromolecule delivery. Discuss potential applications in treating specific skin diseases or conditions and mention any ongoing or future clinical trials exploring these methods.

<Response to Reviewer#1’s specific comment#6>

Based on the reviewer’s comment, we added the following description in the revised manuscript.

From this section, we observed that ItP technology enabled the intradermal delivery of various macromolecules. Most of these applications showed effectiveness against skin-associated diseases, such as ItP of IL-10 for atopic dermatitis, STAT3 siRNA for skin cancer, and TNF-α drug etanercept for psoriasis. However, the application of ItP for intradermal delivery of macromolecules remains at the pre-clinical stage and has not yet been clinically used. In addition to macromolecules, ItP of small molecule drugs is investigated for the treatment of a wide range of skin conditions. This manuscript does not review this context as we focus on ItP-mediated intradermal delivery of macromolecules and mRNA vaccination. For the application of ItP in dermato-cosmetic and aesthetic sciences, interested readers can follow the review with reference [73]. (Pages ##5-6, Lines ##191-200 in the revised manuscript)

Reference:

[73] Liatsopoulou, A.; Varvaresou, A.; Mellou, F.; Protopapa, E. Iontophoresis in dermal delivery. A review of applications in dermato-cosmetic and aesthetic sciences. Int. J. of Cosmet. Sci. 2022, 45, 117-132.  doi: https://doi.org/10.1111/ics.12824.

<Reviewer#1’s specific comment#7>

Briefly explain the mechanisms through which ItP facilitates the delivery of various macromolecules, including the role of electric currents, electrostatic repulsion, and other key processes that enable intradermal delivery.

<Response to Reviewer#1’s specific comment#7>

The ItP-mediated mechanisms in which ItP facilitates the delivery of various macromolecules, including the role of electric currents, such as electrostatic repulsion, electroosmosis, and the biological effects of electric current, have already been discussed in section 3. Additionally, these mechanisms are briefly explained in the figure 1. And we already mentioned this point in the Response to Reviewer#1’s specific comment#5. We will be grateful if the reviewer would check it.

<Reviewer#1’s specific comment#8>

Please, ensure that specialized terms and abbreviations are defined or explained, especially for readers who may not be familiar with specific macromolecules or medical terms used in this section.

<Response to Reviewer#1’s specific comment#8>

Based on the reviewer’s comment, to ensure the specialized terms and abbreviations we added the following section in the revised manuscript.

Abbreviations: mRNA, Messenger RNA; ItP, iontophoresis; FDA, food and drug administration; UTR, untranslated region; APCs, antigen-presenting cells; WEC, weak electric current; mA, milli ampere; siRNA, short interfering RNA; IL-10, interleukin-10; STAT3, signal transducer and activator of transcription 3; HSP, heat shock protein; CCl4, carbon tetrachloride; Pdx, pancreatic and duodenal homeobox;  SOD, superoxide dismutase; UV, ultraviolet; ROS, reactive oxygen species; TNF-α, tumor necrosis factor-α;  IMQ, imiquimod; μg, microgram; NK, natural killer; EGFR, epidermal growth factor receptor; SCC, squamous cell carcinoma; SC, stratum corneum; PKC, protein kinase C, ISMB, in-stem molecular beacon; GFP, green fluorescent protein; CLSM, confocal laser scanning microscopy; FITC, fluorescein isothiocyanate; MW, molecular weight; iTRAQ, isobaric tags for relative and absolute quantification; PEG, polyethylene glycol; nm, nanometer, mV, milli volt; TEM, transmission electron microscopy, LCs, Langerhans cells; TAA, tumor-associated antigen; IFN, interferon; ODN, oligodeoxyribonucleotides; EMA, European Medical Agency  (Page ##14, Lines ##519-530 in the revised manuscript)

<Reviewer#1’s specific comment#9>

Please, emphasize the clinical relevance of these mechanisms for the delivery of naked mRNA vaccines. Explain how these mechanisms can enhance vaccine effectiveness and improve immune responses.

<Response to Reviewer#1’s specific comment#9>

The application of ItP for intradermal delivery of macromolecules and naked mRNA remains in the pre-clinical stage. Therefore, the authors cannot explain specifically the clinical relevance of ItP-associated mechanisms for delivering naked mRNA. We discussed the clinical implication of ItP and their relevance in response to the reviewer's comment #12 and added a new section as “6. Clinical implication of ItP and associated trials” in the revised manuscript. Regarding how ItP-associated mechanisms can enhance vaccine effectiveness and improve immune responses, we already discussed these contexts in section 4. We would be grateful if the reviewer would check it.

<Reviewer#1’s specific comment#10>

Provide more context and examples of the biological mechanisms involved in intradermal delivery, particularly as they relate to naked mRNA vaccines. This will help readers grasp the significance of these mechanisms.

<Response to Reviewer#1’s specific comment#10>

Based on the review's comment, we expand our discussion regarding the biological mechanisms involved in the intradermal delivery as follows:

The skin exhibits a distinct barrier function that restricts the entry of extraneous agents into the body. Skin barrier function is attributed to SC, which is composed of highly organized corneocytes surrounded by a lipid matrix containing ceramides, cholesterol, and fatty acids [82]. Underneath the SC is the viable epidermis, followed by the dermis layer. Topically applied drugs or macromolecules have three possible routes to overcome SC [83,84]. For example, the intracellular route through the cells, the intercellular or paracellular route between the cells, and the follicular or trans-appendageal route following hair follicles. Small lipophilic molecules can penetrate SC following the intracellular route. On the other hand, hydrophilic polar molecules are favorable for paracellular or trans-appendageal routes. However, penetration following these routes typically depends on the properties of drug molecules, the vehicle used in the formulation, and the types of drug delivery systems. (Page ##7, Lines ##232-243 in the revised manuscript)

Reference:

[82] Hatta, I.; Nakazawa, H.; Ohta, N.; Uchino, T.; Yanase, K. Stratum corneum function: a structural study with dynamic synchrotron X-ray diffraction dxperiments. J. Oleo Sci. 2021, 70, 1181–1199, doi: https://doi.org/10.5650/jos.ess21159. 

[83] Yu, Y. Q.; Yang, X.; Wu, X. F.; Fan, Y. B. Enhancing permeation of drug molecules across the skin via delivery in nanocarriers: novel strategies for effective transdermal applications. Front. Bioeng. Biotechnol. 2021, 9, 646554, doi: https://doi.org/10.3389/fbioe.2021.646554.

[84] Tran, T.N.T. Cutaneous Drug Delivery: An Update. J. Investig. Dermatol. Symp. Proc.  2013, 16, S67–S69, doi: https://doi.org/10.1038/jidsymp.2013.28

<Reviewer#1’s specific comment#11>

Please, elaborate on the concept of endosomal escape, explaining why it's crucial for the success of mRNA vaccines and how ItP can facilitate this process.

Based on the reviewer’s comment, we provided additional explanations in the revised manuscript as follows:

Moreover, during cellular entry of mRNA molecules by endocytosis, they would be trapped in endosomes. Endocytosis pathways typically consist of early endosomes, recycling endosomes, multivesicular bodies, late endosomes, and lysosomes [99]. Following this pathway, mRNA molecules recycle to the cell surface or fuse with lysosomes, which contain degradation enzymes. Thus, endosomal escape is essential to access mRNA molecules into the cytoplasm. (Page ##10, Lines ##348-353 in the revised manuscript)

WEC-induced cytoplasmic delivery occurs by unique endocytosis that forms leaky endosomes via localization of ceremide that produces channels or pores in the endosomes. Since ribonucleases degrade naked mRNA in the extracellular environment and inside endosomes, a prolonged stay in such an environment reduces the mRNA vaccine activity. Therefore, ItP-mediated rapid cellular uptake and subsequent escape from leaky endosomes will facilitate effective cytoplasmic delivery of mRNA. (Page ##10, Lines ##357-363 in the revised manuscript)

Reference:

[99] Varkouhi, A.K.; Scholte, M.; Storm, G.; Haisma, H.J. Endosomal escape pathways for delivery of biologicals. J. Control Release 2011, 151, 220–228, doi: https://doi.org/10.1016/j.jconrel.2010.11.004.

<Reviewer#1’s specific comment#12>

Please, discuss the potential clinical implications of using ItP for mRNA vaccine delivery. Explain how these advantages can lead to improved vaccine effectiveness and patient compliance, especially in comparison to conventional delivery methods.

<Response to Reviewer#1’s specific comment#12>

According to the reviewer’s comment, we added a new section in the revised manuscript describing the clinical implications of ItP as follows:

  1. Clinical implication of ItP and associated trials

ItP has shown effectiveness in the intradermal delivery of a number of small molecule drugs against various skin disorders. Recently, ItP-mediated intradermal delivery of macromolecules and vaccines for the treatment of skin cancers has been investigated [49,50]. In addition to the dermal and intradermal application, ocular ItP and ItP on the internal organs have been explored [112,44]. Transpapillary ItP-mediated delivery of chemotherapeutic drugs to specific cancer tissue to minimize systemic toxicity has been investigated [113]. Despite this wide range of applications, ItP is not clinically advanced, and most of its applications remain at the laboratory level. To date, FDA approved ItP for dermal analgesia, management of post-operative pain, and migraine [114, 45]. After surgery, patients suffer from post-operative pain. Conventionally, intravenously administrated patient-controlled analgesia is used to manage post-operative pain. IONSYS® is a needle-free fentanyl ItP patch that offers patient-controlled on demand fentanyl delivery to the post-operative patient and is used as an alternative to conventional pain management system [114]. Zecuity® was a sumatriptan ItP patch used for the management of migraine. Sumatriptan is a hydrophilic drug that acts as an anti-migraine agent [45]. Oral formulation of sumatriptan exhibits poor bioavailability. Moreover, migraine attack is associated with nausea and vomiting, which makes oral administration unstable. Thus, ItP offers noninvasive transdermal delivery of sumatriptan that improves patient compliance. LidoSite® was an ItP patch that contained lidocaine and epinephrine [114]. After ItP-mediated combined delivery, epinephrine reduced the blood flow at the delivery site to increase the local concentration of lidocaine. Thus, the combined effect causes rapid dermal analgesia. Although FDA has approved these ItP patches, these products have been discontinued from the market because of some issues, such as the availability of the ItP device, safety, and lack of sales [114,45].

Recently, several clinical trials associated with ItP have been reported. For example, ItP of dexamethasone and gel lidocaine for the treatment of lateral epicondylitis [115], ItP of methotrexate against palmar psoriasis [116],  ItP-mediated transdermal delivery of vitamin C for the treatment of photoaging skin [117], ItP of neostigmine/glycopyrrolate to initiate bowel evacuation in patients with spinal cord injury [118]. These clinical trials showed promising results for treating corresponding diseases and ensuring the ItP approach is safe and effective. In recent years, the field of biological macromolecular drugs is expanding. Among them, mRNA vaccine development draws significant attention. Conventional hypodermic injection is used to administer the vaccines. We discussed the limitations of such administration earlier in this manuscript, which need to be improved. ItP of naked mRNA vaccine would be an alternative to conventional delivery methods. Furthermore, painless administration and easy application of ItP will enhance patient compliance. (Pages ##12-13, Lines ##440-477 in the revised manuscript)

Reference:

[112] Dong, W.; Pu, N.; Li, S.; Wang, Y.; Tao, Y. Application of iontophoresis in ophthalmic practice: an innovative strategy to deliver drugs into the eye. Drug Deliv. 2023, 30, doi: https://doi.org/10.1080/10717544.2023.2165736.

[113] Gadag, S.; Narayan, R.; Sabhahit, J.N.; Hari, G.; Nayak, Y.; Pai, K.S.R.; Garg, S.; Nayak, U.Y. Transpapillary iontophoretic delivery of resveratrol loaded transfersomes for localized delivery to breast cancer. Biomater. Adv. 2022, 140, 213085, doi: https://doi.org/10.1016/j.bioadv.2022.213085.  

[114] Bakshi, P.; Vora, D.; Hemmady, K.; Banga, A.K. Iontophoretic skin delivery systems: success and failures. Int. J.  of Pharm. 2020, 586, 119584, doi: https://doi.org/10.1016/j.ijpharm.2020.119584.

[115] da Luz, D.C.; de Borba, Y.; Ravanello, E.M.; Daitx, R.B.; Döhnert, M.B. Iontophoresis in lateral epicondylitis: a randomized, double-Blind clinical trial. J. Shoulder Elbow. Surg. 2019, 28, 1743–1749, doi: https://doi.org/10.1016/j.jse.2019.05.020.  

[116] Andanooru Chandrappa, N.K.; Channakeshavaiah Ravikumar, B.; Rangegowda, S.M. Iontophoretic Delivery of Methotrexate in the Treatment of Palmar Psoriasis: A Randomised Controlled Study. The Australas. J.  Dermatol. 2020, 61, 140–146, doi: https://doi.org/10.1111/ajd.13228.

[117] Correia, G.; Magina, S. Efficacy of topical vitamin c in melasma and photoaging: a systematic review. J. Cosmet. Dermatol.2023, 7, 1938-1945 doi: https://doi.org/10.1111/jocd.15748.

[118] Korsten, M.A.; Lyons, B.L.; Radulovic, M.; Cummings, T.M.; Sikka, G.; Singh, K.; Hobson, J.C.; Sabiev, A.; Spungen, A.M.; Bauman, W.A. Delivery of neostigmine and glycopyrrolate by iontophoresis: a nonrandomized study in individuals with spinal cord injury. Spinal Cord 2017, 56, 212–217, doi: https://doi.org/10.1038/s41393-017-0018-2.

<Reviewer#1’s specific comment#13>

Please, consider mentioning any potential synergies or limitations when combining ItP with other delivery methods, as briefly touched upon in the section about improved transdermal delivery of rabies vaccines. This could provide a more comprehensive view of ItP's role in vaccine delivery.

<Response to Reviewer#1’s specific comment#13>

As the reviewer pointed out about the potential synergies or limitations of ItP combined with other methods, we amended our text as follows:

ItP, in combination with other permeation methods, did not report any potential adverse effects. However, the changes in the skin structure, irritation, and erythema associated with the combined application of ItP with each method need to be investigated. Additionally, the combined application will be complicated and expensive relative to the single application of ItP. (Page ##12, Lines ##434-439 in the revised manuscript)

<Reviewer#1’s specific comment#14>

Please, expand on the future perspectives of ItP technology in the context of intradermal vaccination. Discuss ongoing research or potential developments that could further enhance the application of ItP for vaccine delivery.

<Response to Reviewer#1’s specific comment#13>

Based on the reviewer's comment, we expand the future perspective and discuss the ongoing research and developments as follows:

Several challenges need to be overcome to extend the application of ItP technology for intradermal vaccination. The delivery efficacy of mRNA and associated immune responses need to improve. ItP patches equipped with mRNA vaccines and cost-effective smart ItP devices need to be developed. Due to the rapid development of microelectronics, recently, cell phone-used remote-controllable dosage management ItP system, self-powered ItP patches, eco-friendly energy used ItP patches have been reported [121-123]. These developments are expected to extend the applicability of ItP. Furthermore, for clinical translation of ItP technology, the use of human-relevant models such as human skin explants, tissue-engineered skin models, and human-based tissues grafted onto mice needs to increase [124-126]. More research and clinical trials will improve the application of ItP for vaccine delivery in the future. (Page ##14, Lines ##508-518 in the revised manuscript)

Reference:

[121] Mori, K.; Yamazaki, K.; Takei, C.; Takeshi O.; Takeuchi, I.; Miyaji, K.; Hiroaki T.; Shoko I.; Kenji S. Remote-controllable dosage management through a wearable iontophoretic patch utilizing a cell phone. J. Control. Release 2023, 355, 1–6, doi: https://doi.org/10.1016/j.jconrel.2023.01.046.  

[122] Wu, C.; Jiang, P.; Li, W.; Guo, H.; Wang, J.; Chen, J.; Prausnitz, M.R.; Wang, Z.L. Self-powered iontophoretic transdermal drug delivery system driven and regulated by biomechanical motions. Adv. Funct. Mater. 2019, 30, 1907378, doi: https://doi.org/10.1002/adfm.201907378.

[123] Lee, J.; Kwon, K.; Kim, M.; Min, J.; Hwang, N.S.; Kim, W. Transdermal iontophoresis patch with reverse electrodialysis. Drug Deliv. 2017, 24, 701–706, doi: https://doi.org/10.1080/10717544.2017.1282555.

[124] Shannon, J.; Kirchner, S.; Zhang, J. Human skin explant preparation and culture. Bio. Protoc. 2022, 12, doi: https://doi.org/10.21769/bioprotoc.4514.

[125] Suhail, S.; Sardashti, N.; Jaiswal, D.; Rudraiah, S.; Misra, M.; Kumbar, S.G. Engineered skin tissue equivalents for product evaluation and therapeutic applications. Biotechnol. J. 2019, 14, e1900022, doi: https://doi.org/10.1002/biot.201900022.

[126] Kenney, L.L.; Shultz, L.D.; Greiner, D.L.; Brehm, M.A. Humanized mouse models for transplant immunology. Am. J. Transpl. 2015, 16, 389–397, doi: https://doi.org/10.1111/ajt.13520.

.

We wish to express our sincere thanks to Reviewer#1.

Reviewer 2 Report

Comments and Suggestions for Authors

Comments on the Quality of English Language

Good. Minor edits.

Author Response

To Reviewer#2

< Reviewer#2’s comment>

The authors reviewed the current status of using iontophoresis approach intradermal delivery of naked mRNA vaccines. It is well written and informative. The authors have worked in the field of iontophoresis delivery approach. It would be beneficial to the readers if the authors can describe the instrument that is used for the iontophoresis delivery. Also, the stability and storage of the naked mRNA vaccine vs mRNA-LNP delivery system.

<Response to Reviewer#2’s comment>

Thank you for reviewing our manuscript. We express our appreciation to Reviewer#2 for evaluating our works and valuable comments. As the reviewer pointed out about the ItP instrument and stability of mRNA vaccines, we added an explanation in the revised manuscript as follows:

  1. ItP instrument, stability and storage of mRNA vaccines

A primary ItP instrument typically contains a power source, an electric controller, two electrodes (e.g., anode and cathode), and a drug reservoir. Two electrodes are placed on the skin surface during ItP application to complete the circuit. The drug reservoir is embedded with the active electrode, while the return electrode contains counter ions. Earlier, the ItP device was large and required an external power source. However, due to the rapid development of the technology, the ItP device has become updated. Recently, self-powered, smart ItP devices have been introduced and are commercially available. Tula® System, Hidrex PSP1000, Idromed 5 PS, and Fischer Galvanic MD-2 are some FDA-approved ItP devices available in the market [115,45].

mRNA vaccines showed remarkable success in combating the COVID-19 pandemic. Despite the effectiveness and safety of mRNA vaccines, their stability is not clearly understood. Several factors, such as the vehicle used in the formulation, pH, and temperature, can affect the stability of the mRNA vaccines [119]. Pfizer/BioNTech and Moderna used lipid nanoparticle-formulated mRNA vaccines. According to the European Medical Agency (EMA), both vaccines are stable in frozen condition for up to 6 months at −25 °C, up to 30 days at refrigerator temperature (4°C), and up to 6 h at room temperature [119]. Regarding the storage conditions of naked mRNA, no specific guideline is found. However, Jones et al. found that freeze-dried with trehalose permitted stable storage of functional RNA at 4°C for up to 10 months [120]. (Page s##13-14, Lines ##478-497 in the revised manuscript)

Reference:

[114] Bakshi, P.; Vora, D.; Hemmady, K.; Banga, A.K. Iontophoretic skin delivery systems: success and failures. Int. J.  of Pharm. 2020, 586, 119584, doi: https://doi.org/10.1016/j.ijpharm.2020.119584.

[119] Uddin, M.N.; Roni, M.A. Challenges of storage and stability of mRNA-based COVID-19 vaccines. Vaccines 2021, 9, 1033, doi: https://doi.org/10.3390/vaccines9091033.

[120] Jones, K.L.; Drane, D.; Gowans, E.J. Long-term storage of DNA-free RNA for use in vaccine studies. Bio. Techniques. 2007, 43, 675–681, doi: https://doi.org/10.2144/000112593.

< Reviewer#2’s suggestions>

Minor English edits are suggested:

Line 44: “so that the rate and expression ability an mRNA vaccine's antigen”

Should be “so that the rate and expression ability of an mRNA vaccine's antigen”

Line 51: “vaccines also exhibit have some limitations, specifically such as, they exhibit high molecular weights”

Line 163: “However, it may not be clear just how these molecules are able to permeate” Line 280: “In the present study, appealing features of ItP were demonstrated that it will be”

<Response to Reviewer#2’s suggestions>

According to the reviewer’s suggestions, we accepted all above English edits in the  revised manuscript.

We wish to express our sincere thanks to Reviewer#2

Reviewer 3 Report

Comments and Suggestions for Authors

The title of the submission is " Intradermal delivery of naked mRNA vaccines by iontophoresis" suggests that a very exciting update, overview or critique will follow. The paper instead gives an account of the delivery of all sorts pharmaceuticals and other substances by iontophoresis but little about the delivery of naked DNA vaccines with or without the use of iontophoresis. These include irrelevant accounts of siRNA and CpG (DNA) oligonucleotide adjuvant.

Reference 31 describes the delivery of a single epitope  via intranodal injection of  naked mRNA and immune responses in TCR transgenic mice and tumour rejection rejection studies with the wild type mice (with the right MHC for the epitope). This is a promising study but raises questions about the ability to deliver a more complex vaccine for human application.

Reference 32 is a wide ranging clinical review of mRNA anti-cancer vaccine studies . It does reference  two studies in which intranodal delivery of naked cancer RNA have been used in trails but shouldn't the strategy of the delivery and the results of the specific  studies be the subject of this paper?

Reference 33 describes the delivery of of naked mRNA encoding a fluorescent protein (18 kDa) into excised pig skin and that the expression was the same as a lipid formulation. Shouldn't this paper discuss this and present the important messages of what can be expressed and other factors such as RNA dose and the amount of expression?

Reference 93 does in fact the study the transdermal delivery of a naked mRNA encoding a minimal epitope by in a mouse model. Inhibition of tumour growth was described but there were no controls for an adjuvant effect of the mRNA (critical) or the treatment and no measurement of immune responses.

It is suggested that a critique on naked mRNA delivery and the use of  iontophoresis for vaccines would be useful if it focussed on the progress (or it seems the lack thereof), the techniques and strategies and what needs to be accomplished. Besides the use in cancer immunotherapy covered here the use for infectious disease and mass vaccination where multiple deliveries create logistic problems should be covered.

Author Response

To Reviewer#3

Thank you for reviewing our manuscript. We wish to express our appreciation to Reviewer#3 for valuable comments.

< Reviewer#3’s comment#1>

The title of the submission is " Intradermal delivery of naked mRNA vaccines by iontophoresis" suggests that a very exciting update, overview or critique will follow. The paper instead gives an account of the delivery of all sorts pharmaceuticals and other substances by iontophoresis but little about the delivery of naked DNA vaccines with or without the use of iontophoresis. These include irrelevant accounts of siRNA and CpG (DNA) oligonucleotide adjuvant.

<Response to Reviewer#3’s comment #1>

We agree with Reviewer#3 that the manuscript discussed iontophoresis-mediated intradermal delivery of various macromolecules but little about the DNA vaccines. Moreover, it also addressed the siRNA and  CpG (DNA) oligonucleotide adjuvant. Although it is through that iontophoresis is suitable for the transdermal delivery of small molecule drugs, we and some other research groups explore this technology for intradermal delivery of macromolecules. To date, limited studies have been conducted regarding the iontophoresis-based delivery of naked DNA and mRNA vaccines. Therefore, we could not add more contexts, considering the title of the manuscript. Furthermore, we discussed the siRNA and CpG (DNA) oligonucleotide adjuvant as examples to provide an insight that iontophoresis technology will also be applicable for naked mRNA vaccinations.

< Reviewer#3’s comment#2>

Reference 31 describes the delivery of a single epitope via intranodal injection of naked mRNA and immune responses in TCR transgenic mice and tumour rejection rejection studies with the wild type mice (with the right MHC for the epitope). This is a promising study but raises questions about the ability to deliver a more complex vaccine for human application.

<Response to Reviewer#3’s comment #2 >

Although the authors of this study (reference 31(35)) did not discuss their findings for human application or clinical translation. However, this study is the preclinical proof of concept that intranodal immunization with naked antigen-encoding RNA is a feasible, safe, and powerful approach for antitumoral vaccination. We referred to this study in our manuscript to indicate the feasibility of the application of naked mRNA. Based on the reviewer's comment, we amended our description as follows:

Recently, intranodal and intradermal administration of naked mRNA has been investigated [35,36]. Sebastian et al. reported the potent immune response upon delivery of a single epitope by intranodal injection of naked mRNA using mice models [35]. (Page ##3, Lines ##90-93 in the revised manuscript)

Although these findings did not discuss some points (e.g., clinical application, doses of mRNA, and expression amount), they indicate the proof of concept and feasibility of the application of naked mRNA. (Page ##3, Lines ##96-98 in the revised manuscript)

Reference:

[35] Kreiter, S.; Selmi, A.; Diken, M.; Koslowski, M.; Britten, C.M.; Huber, C.; Türeci, Ö.; Sahin, U. Intranodal vac-cination with naked antigen-encoding RNA elicits potent prophylactic and therapeutic antitumoral immunity. Cancer Res. 2010, 70, 9031–9040, doi: https://doi.org/10.1158/0008-5472.can-10-0699.

< Reviewer#3’s comment#3>

Reference 32 is a wide ranging clinical review of mRNA anti- cancer vaccine studies . It does reference two studies in which intranodal delivery of naked cancer RNA have been used in trails but shouldn't the strategy of the delivery and the results of the specific studies be the subject of this paper?

<Response to Reviewer#3’s comment #3 >

As Reviewer pointed out the reference 32, we removed it from the revised manuscript.

< Reviewer#3’s comment#4>

Reference 33 describes the delivery of of naked mRNA encoding a fluorescent protein (18 kDa) into excised pig skin and that the expression was the same as a lipid formulation. Shouldn't this paper discuss this and present the important messages of what can be expressed and other factors such as RNA dose and the amount of expression?

 <Response to Reviewer#3’s comment # 4 >

In response to the Reviewer's comment, we amended the text as follows:

Furthermore, Sonia et al. investigated the intradermal delivery of naked mRNA encoding a fluorescent protein into excised pig skin [36]. They found that the intradermal delivery of naked mRNA resulted in protein expression. Although these findings did not discuss some points (e.g., clinical application, doses of mRNA, and expression amount), they indicate the proof of concept and feasibility of the application of naked mRNA. (Page ##3, Lines ##93-98 in the revised manuscript)

[36]. Golombek, S.; Pilz, M.; Steinle, H.; Efrat Kochba; Levin, Y.; Lunter, D.J.; Schlensak, C.; Wendel, H.P.; Avci‐Adali, M. Intradermal delivery of synthetic mRNA using hollow microneedles for efficient and rapid production of exogenous proteins in skin. Mol. Ther. Nucleic acids 2018, 11, 382–392, doi: https://doi.org/10.1016/j.omtn.2018.03.005.

< Reviewer#3’s comment#5>

Reference 93 does in fact the study the transdermal delivery of a naked mRNA encoding a minimal epitope by in a mouse model. Inhibition of tumor growth was described but there were no controls for an adjuvant effect of the mRNA (critical) or the treatment and no measurement of immune responses.

<Response to Reviewer#3’s comment # 5 >

According to Reviewer#3’s comment#5, we added the description about an adjuvant effect of the mRNA regarding Reference 93 (102) as follows:

The tumor growth inhibition by ItP of mRNA might be due to not only antigen derived immune response but also an adjuvant effect. Therefore, the ItP treatment of mRNA encoding non-related peptide should be examined in the future. (Page##11, Lines ## 398-401 in the revised manuscript)

Regarding immune responses after ItP of mRNA, we already mentioned in the original manuscript as “To confirm the immune responses, the authors found that there was an elevation in mRNA expression levels of various cytokines, such as IFN-γ, TNF-α, and IL-12b, in both the skin and tumor tissues in the ItP-vaccinated group compared to the non-vaccinated group. The elevated cytokine levels contribute to the suppression of tumor growth. Furthermore, the infiltration of cytotoxic CD8+ T cells was also found in the tumor tissue, which is responsible for tumor clearance.” We will be grateful if the Reviewer#3 would check it.

< Reviewer#3’s comment#6>

It is suggested that a critique on naked mRNA delivery and the use of iontophoresis for vaccines would be useful if it focussed on the progress (or it seems the lack thereof), the techniques and strategies and what needs to be accomplished. Besides the use in cancer immunotherapy covered here the use for infectious disease and mass vaccination where multiple deliveries create logistic problems should be covered.

<Response to Reviewer#3’s comment # 6>

In recent years, the field of biological macromolecular drugs is expanding. Although ItP is typically applied for small molecule drugs, it has garnered significant interest for intradermal vaccinations. Moreover, to improve the application system, the ItP device is upgraded day by day. We expand our discussion about the recent development and future needs mentioned below. The iontophoresis-based intradermal delivery of vaccines is advantageous for both cancer immunotherapy and infection disease. Although the reviewer thinks multiple deliveries create logistic problems, as ItP is a delivery technology, it has no potential variations regarding intradermal vaccination for cancer immunotherapy or infectious diseases.

Several challenges need to be overcome to extend the application of ItP technology for intradermal vaccination. The delivery efficacy of mRNA and associated immune responses need to improve. ItP patches equipped with mRNA vaccines and cost-effective smart ItP devices need to be developed. Due to the rapid development of microelectronics, recently, cell phone-used remote-controllable dosage management ItP system, self-powered ItP patches, eco-friendly energy used ItP patches have been reported [121-123]. These developments are expected to extend the applicability of ItP. Furthermore, for clinical translation of ItP technology, the use of human-relevant models such as human skin explants, tissue-engineered skin models, and human-based tissues grafted onto mice needs to increase [124-126]. More research and clinical trials will improve the application of ItP for vaccine delivery in the future. (Page ##14, Lines ##508-518 in the revised manuscript)

Reference:

[121] Mori, K.; Yamazaki, K.; Takei, C.; Takeshi O.; Takeuchi, I.; Miyaji, K.; Hiroaki T.; Shoko I.; Kenji S. Remote-controllable dosage management through a wearable iontophoretic patch utilizing a cell phone. J. Control. Release 2023, 355, 1–6, doi: https://doi.org/10.1016/j.jconrel.2023.01.046. 

[122] Wu, C.; Jiang, P.; Li, W.; Guo, H.; Wang, J.; Chen, J.; Prausnitz, M.R.; Wang, Z.L. Self-powered iontophoretic transdermal drug delivery system driven and regulated by biomechanical motions. Adv. Funct. Mater. 2019, 30, 1907378, doi: https://doi.org/10.1002/adfm.201907378.

[123] Lee, J.; Kwon, K.; Kim, M.; Min, J.; Hwang, N.S.; Kim, W. Transdermal iontophoresis patch with reverse electrodialysis. Drug Deliv. 2017, 24, 701–706, doi: https://doi.org/10.1080/10717544.2017.1282555.

[124] Shannon, J.; Kirchner, S.; Zhang, J. Human skin explant preparation and culture. Bio. Protoc. 2022, 12, doi: https://doi.org/10.21769/bioprotoc.4514.

[125] Suhail, S.; Sardashti, N.; Jaiswal, D.; Rudraiah, S.; Misra, M.; Kumbar, S.G. Engineered skin tissue equivalents for product evaluation and therapeutic applications. Biotechnol. J. 2019, 14, e1900022, doi: https://doi.org/10.1002/biot.201900022.

[126] Kenney, L.L.; Shultz, L.D.; Greiner, D.L.; Brehm, M.A. Humanized mouse models for transplant immunology. Am. J. Transpl. 2015, 16, 389–397, doi: https://doi.org/10.1111/ajt.13520.

We wish to express our sincere thanks to Reviewer#3.

Round 2

Reviewer 1 Report

Comments and Suggestions for Authors

The authors have made sufficient corrections for their work to be published in Pharmaceutics.

Reviewer 3 Report

Comments and Suggestions for Authors

Revision gives a better perspective